# Chronic Kidney Disease—An Underestimated Risk Factor for Antimicrobial Resistance in Patients with Urinary Tract Infections

**DOI:** 10.3390/biomedicines10102368

**Published:** 2022-09-22

**Authors:** Ileana Adela Vacaroiu, Elena Cuiban, Bogdan Florin Geavlete, Valeriu Gheorghita, Cristiana David, Cosmin Victor Ene, Catalin Bulai, Gabriela Elena Lupusoru, Mircea Lupusoru, Andra Elena Balcangiu-Stroescu, Larisa Florina Feier, Ioana Sorina Simion, Daniela Radulescu

**Affiliations:** 1Department of Nephrology, Carol Davila University of Medicine and Pharmacy, 020021 Bucharest, Romania; 2Department of Nephrology, Sfantul Ioan Clinical Emergency Hospital, 042122 Bucharest, Romania; 3Department of Urology, Carol Davila University of Medicine and Pharmacy, 020021 Bucharest, Romania; 4Department of Urology, Sfantul Ioan Clinical Emergency Hospital, 042122 Bucharest, Romania; 5Faculty of Medicine, Carol Davila University of Medicine and Pharmacy, 020021 Bucharest, Romania; 6Prof. Dr. Agrippa Ionescu Clinical Emergency Hospital, 011356 Bucharest, Romania; 7Department of Nephrology, Fundeni Clinical Institute, 022328 Bucharest, Romania; 8Department of Physiology, Carol Davila University of Medicine and Pharmacy, 020021 Bucharest, Romania; 9Discipline of Physiology, Faculty of Dental Medicine, Carol Davila University of Medicine and Pharmacy, 020021 Bucharest, Romania

**Keywords:** chronic kidney disease, hemodialysis, urinary tract infections, infection, antimicrobial resistance, multiple drug resistance

## Abstract

(1) Background: Chronic kidney disease (CKD), as well as antimicrobial resistance (AMR) represent major global health problems, with important social and economic implications. It was reported that CKD is a risk factor for antimicrobial resistance, but evidence is scarce. In addition, CKD is recognized to be a risk factor for complicated urinary tract infections (UTIs). (2) Methods: We conducted an observational study on 564 adult in-hospital patients diagnosed with urinary tract infections. The aim of the study was to identify the risk factors for AMR, as well as multiple drug resistance (MDR) and the implicated resistance patterns. (3) Results: The mean age was 68.63 ± 17.2 years. The most frequently isolated uropathogens were *Escherichia coli* strains (68.3%) followed by *Klebsiella* species (spp. (11.2%). In 307 cases (54.4%)), the UTIs were determined by antibiotic-resistant bacteria (ARBs) and 169 cases (30%) were UTIs with MDR strains. Increased age (≥65) OR 2.156 (95% CI: 1.404–3.311), upper urinary tract obstruction OR 1.666 (1.083–2.564), indwelling urinary catheters OR 6.066 (3.919–9.390), chronic kidney disease OR 2.696 (1.832–3.969), chronic hemodialysis OR 4.955 (1.828–13.435) and active malignancies OR 1.962 (1.087–3.540) were independent risk factors for MDR UTIs. In a multivariate logistic regression model, only indwelling urinary catheters (OR 5.388, 95% CI: 3.294–8.814, *p* < 0.001), CKD (OR 1.779, 95% CI: 1.153–2.745, *p* = 0.009) and chronic hemodialysis (OR 4.068, 95% 1.413–11.715, *p* = 0.009) were risk factors for UTIs caused by MDR uropathogens. (4) Conclusions: CKD is an important risk factor for overall antimicrobial resistance, but also for multiple-drug resistance.

## 1. Introduction

Chronic kidney disease (CKD) represents a global health problem, with a prevalence of more than 10% in the adult world population, with important epidemiological differences by geographic areas [1]. To note that CKD is one of the top 20 causes of death globally, progressing from the 17th leading cause of death in 1990 to the 12th leading cause of death in 2017 according to the Global Burden of Disease (GBD) study [2]. Infections are an important cause of hospitalization in CKD patients of all stages, with graded association of a higher risk of infection with more advanced stages of disease [3,4,5,6] and a major cause of mortality in end-stage kidney disease (ESKD), being exceeded only by cardiovascular causes of death [4,7].

Urinary tract infections (UTIs) are one of the most common type of community and hospital-acquired bacterial infections, with heterogenous clinical expressions from asymptomatic bacteriuria or uncomplicated cystitis to life-threatening forms of urosepsis, evolution and outcomes being influenced by the host risk factors (male sex, pregnancy, immune deficiencies, kidney diseases, diabetes mellitus, obesity, previous urinary tract infections, indwelling catheters, urinary tract abnormalities, etc.) and pathogen-specific risk factors (increased refraction against host immune responses, invasiveness, adhesins, biofilm and intracellular colonies’ formation, escape mechanisms such as capsule formation, and antibiotic resistance) [8,9].

It is generally accepted that CKD is a risk factor for complicated UTIs [8,9] and the clinical evolution and outcomes of UTIs in this specific population are influenced by higher levels of comorbidity, impaired immunity in people with CKD, and equally important, by the increasing prevalence of UTIs with antibiotic-resistant uropathogens [10,11].

Furthermore, antimicrobial resistance (AMR) represents a major and growing worldwide public health issue and one of the top ten global health threats as stated by World Health Organization [12], but also a social and economic problem, with a progressively growing burden. The major importance of the topic is highlighted by numerous international and local public health action plans and initiatives [13,14,15] and guidelines’ recommendations to initiate empirical treatment according to local antibiotic resistance patterns [8].

Even if it was reported that CKD, especially ESKD, is a risk factor for antimicrobial resistance [16,17,18,19], the evidence is inconsistent and further research is needed.

## 2. Materials and Methods

We conducted a single-center observational study of adult in-hospital patients admitted to the Nephrology Department of “Sf. Ioan” Emergency Clinical Hospital, Bucharest during a period of 24 months (between 1 January 2019 and 31 December 2020) with a diagnosis of urinary tract infection at discharge according to the International Classification of Diseases, 10th Revision, Clinical Modification (ICD-10-CM), protocol code 20960 approved on 9 June 2021 by the Institutional Ethics Committee of Sfantul Ioan Emergency Clinical Hospital. Data were retrieved from the hospital electronic medical records database. A total of 1103 cases were assessed for eligibility, and 564 patients met the inclusion criteria and were further included in the final study analysis.

### 2.1. Inclusion Criteria

Diagnosis of urinary tract infection at discharge according to the ICD-10-CM;

Microbiological confirmation of UTI, namely bacterial growth of ≥10^5^ colony forming units (CFU)/mL of a single isolate or growth ≥10^5^ CFU/mL with mixed growth but one predominant strain;

Urine sample collected during first 48 h of hospitalization.

### 2.2. Exclusion Criteria

Sterile urine culture or urine sample collected after 48 h of hospitalization;

Hospital admission by inter-hospital transfer or less than 30 days after a previous hospital discharge or recent urinary tract instrumentation, in order to exclude hospital-acquired urinary tract infections (HAUTIs).

### 2.3. Variables and Definitions

Included variables were divided into categories: demographics (age, sex); clinical data (infection type classified according to European Association of Urology (EAU) Guideline: cystitis, pyelonephritis, urosepsis, recurrent UTIs, catheter-associated urinary tract infections (CAUTIs) [8], symptoms at the time of presentation, risk factors for complicated UTIs and for antimicrobial resistance; laboratory parameters related to inflammatory response to UTI (white blood cells count (WBC), C reactive protein (CRP), fibrinogen); microbiological parameters; antibiotic treatment and outcomes (infectious and non-infectious complications, including acute kidney injury (AKI), length of stay, all-cause mortality).

Multiple drug resistance (MDR) was defined as resistance to ≥3 antibiotic classes [20]. Intermediate levels of resistance were considered as sensitive as they do not exclude prescription of a specific antibiotic. Pyelonephritis was diagnosed according to EAU Guideline statements, based on clinical presentation: fever >38 °C, chills, lumbar pain, vomiting, nausea, costovertebral angle tenderness, with or without cystitis symptoms and imaging finding of pyelonephritis at ultrasound (US) and/or computed tomography (CT) evaluation [8]. CKD was defined and classified based on estimated glomerular filtration rate (eGFR) criteria according to the Kidney Disease: Improving Global Outcomes (KDIGO) 2012 Clinical Practice Guideline for the Evaluation and Management of Chronic Kidney Disease [21]. AKI diagnosis was established and staged based on serum creatinine criteria according to the KDIGO Clinical Practice Guidelines for Acute Kidney Injury 2012 [22].

### 2.4. Statistical Analysis

Statistical analysis was performed using IBM SPSS Statistics for Windows, Version 25.0 and Microsoft Office Excel/Word 2013. Quantitative variables were tested for distribution using the Shapiro–Wilk test and were expressed as means with standard deviations or medians with inter-percentile range. Independent quantitative variables with non-parametric distribution were tested using the Mann–Whitney U test. Qualitative variables were expressed as absolute number or percentage, and were tested using the Fisher’s Exact Test. Z-tests with Bonferroni correction were performed to further analyze the results obtained in contingency tables. Quantification of associations between qualitative variables was performed using odds ratios (OR) with 95% confidence intervals. Multivariate logistic regression models were used to predict overall antimicrobial resistance and multiple drug resistance. Quantification of prediction in logistic regression models was performed using odds ratios (OR) with 95% confidence intervals.

## 3. Results

A total of 564 patients were included in the final study analysis, after excluding patients who did not met the microbiological criteria and patients considered to have HAUTI. In the study group, the mean age was 68.63 ± 17.2 years, with 391 patients (69.3%) aged 65 and over. Male patients were represented in the proportion of 31.8% (179 patients) and females 68.2% (384 patients). The most common infections were pyelonephritis 436 cases (77.4%), followed by urosepsis 93 cases (16.5%) and cystitis 43 cases (7.6%). A total of 173 (30.7%) were recurrent UTIs and 72 (12.8%) cases were CAUTIs. The mean length of stay was 8.33 *±* 5.82 days (Table 1).

We identified several infectious and non-infectious complications in the study group, which influenced the clinical course and outcomes. A total of 23 patients (4.1%) experienced superinfection, 16 patients (2.8%) developed Clostridioides difficile enterocolitis and 34 patients (6.9%) had concurrent bacterial or viral infections, the most frequent being pneumonia in 51.28% cases. Related to non-infectious complications, 197 patients (34.9%) experienced AKI, among them 129 cases (65.5%) were AKI on CKD. Regarding to AKI severity, 99 cases (50.5%) were stage 3 AKI and 52 patients (9.2%) needed acute renal replacement therapy (RRT) (Appendix A).

Microbiologic profiles: the most frequently isolated uropathogens were *Escherichia coli* strains (68.3%) followed by *Klebsiella* species (spp. (11.2%). In 307 cases (54.4%), the UTIs were determined by antibiotic resistant bacteria (ARBs) and 169 cases (30%) were UTIs with MDR strains, with the highest general and multiple drug resistance in *Pseudomonas* species 31 cases (91.2%) and 23 (67.6%), respectively; *Klebsiella* species 46 (73%) and 30 (47.6%), respectively; and *Enterococcus* species 26 (72.2%) and 18 (50%), respectively (Appendix B). No significant differences in microbiological profiles were observed between CKD and non-CKD patients (Table 2).

Univariate and multivariate logistic regression analysis were used to identify independent risk factors for UTIs with antibiotic resistant uropathogens. Univariate analysis revealed a significant association with antibiotic resistance for increased age OR 2.428 (95% CI: 1.683–3.503), upper urinary tract obstruction OR 1.871 (1.214–2.882), lower urinary tract obstruction OR 1.713 (1.047–2.800), indwelling urinary catheters OR 4.787 (2.905–7.888), chronic kidney disease OR 2.502 (1.780–3.517) and nursing home resident status OR 5.638 (1.26–25.21). In addition, the differences between groups in relation to CKD stage and antibiotic resistance were also significant (*p* = 0.021), namely patients with CKD stage 1 were more frequently associated with UTIs with sensitive strains (4.8% vs. 0.5%), with no other significant differences between groups (Table 3; Figure 1).

In multivariate logistic regression analysis for overall antimicrobial resistance prediction only age over 65, indwelling urinary catheters and the CKD were significant predictive factors for occurrence of UTIs with antibiotic resistant uropathogens, CKD patients having a 1846 times higher risk (95% CI:: 1.273–2.677) of UTIs with antibiotic resistant strains (Table 4; Figure 2).

Furthermore, we analyzed the same risk factors for prediction of UTIs with MDRs uropathogens. Univariate analysis showed that increased age OR 2.156 (95% CI: 1.404–3.311), upper urinary tract obstruction OR 1.666 (1.083–2.564), indwelling urinary catheters OR 6.066 (3.919–9.390), chronic kidney disease OR 2.696 (1.832–3.969), chronic hemodialysis OR 4.955 (1.828–13.435) and active malignancies OR 1.962 (1.087–3.540) were all significantly associated with MDRs UTIs (Table 5; Figure 3).

In a multivariate logistic regression model, only indwelling urinary catheters, CKD and chronic hemodialysis were significant predictive factors for occurrence of UTIs with MDR uropathogens, CKD patients having a 1779 times higher chance (95% CI: (1.153–2.745)) of UTIs with MDR strains, while chronic hemodialysis patients had a 4.068 times higher risk of having MDR UTIs (Table 6; Figure 4).

In our group, the highest resistance rates were observed to fluoroquinolones 37.6%, third generation cephalosporines 32.5%, aminoglycosides 34%, aminopenicillins 28.8% and carbapenems 6.1%.

## 4. Discussion

It is known that the CKD population has an increased susceptibility to infectious events, both community-acquired and hospital-acquired [3,4,23,24]. This could be explained in part by frequent contact with the healthcare system, higher comorbidity index, impaired host defense mechanisms secondary to immune system impairments in both humoral and cellular immune responses, cytokine generation and oxidative stress [3,25,26,27,28]. Specifically, for increased susceptibility to UTIs, some of the potential risk factors are urinary tract abnormalities, disruption of the urinary epithelial barrier or impaired regeneration capacity, impaired bladder voiding, impairment in inflammasome signaling [29], progressive loss of kidney functions associated to decreased levels of urinary secreted molecules, namely antimicrobial peptides such as β-defensin 1, urinary uromodulin–a multifunctional protein also implicated in susceptibility and immune response to UTI [30,31] and other urothelium-secreted antimicrobial substances, such as tissue-type plasminogen activator and urokinase-type plasminogen activator [32]. In addition, genetic polymorphisms of genes that encode receptors implicated in the innate immune response, such as CXC-chemokine receptor type 1 (CXCR1) and Toll-like receptor 4 (TLR4) have been identified as increasing susceptibility to UTIs [27,33].

At the same time, CKD seems to be a risk factor for infections caused by antibiotic-resistant pathogens, but most of the published studies are focused on the risk of MDROs infections in ESKD undergoing dialysis [34,35], with a limited number of studies regarding association of early stages of CKD with antimicrobial resistance. To note that the proportion of CKD patients among hospitalized patients with UTIs varies across studies from 21.47% [10] to 28.6% [36], accounting for 42.4% in our study.

In our study, we found that CKD was associated with a significantly higher risk of UTIs with antibiotic-resistant strains. In the univariate analysis, chronic kidney disease, increased age, upper urinary tract obstruction, lower urinary tract obstruction, indwelling urinary catheters and nursing home resident status were all significantly associated with antibiotic resistance. However, in the multivariate analysis, only CKD, age over 65 and presence of indwelling urinary catheters were associated with overall AMR in hospitalized patients with UTIs. Previous studies also reported an increased risk of UTIs with antibiotic-resistant strains in long-term nursing home residents, elderly [10,17], indwelling urinary catheters [10], urinary incontinence [10] and recurrent UTIs [17].

Regarding multiple drug resistance, Su G. et al., reported an increased risk of MDR organisms in patients with lower eGFR (OR 1.19 for eGFR between 30–60 mL/min/1.73 m^2^ and 1.41 in patients with eGFR below 30 mL/min/m^2^ compared to eGFR 60–104 mL/min/1.73 m^2^), but when MDR was analyzed in CKD patients according to specific sample source, it only observed a trend for increased resistance in patients with UTIs [3]. Another study reported an increased risk of MDR in UTIs patients with CKD (OR = 2.75, *p* < 0.001), confirmed also in multivariable analysis (OR = 3.04, 95% CI: 2.23 to 4.13, *p* < 0.001), thus CKD was an independent predictor for MDR [11]. In our study, chronic kidney disease, chronic hemodialysis, increased age, upper urinary tract obstruction, indwelling urinary catheters and active malignancies were all predictors of MDRs, while in a multivariate regression model only CKD, indwelling urinary catheters, and chronic hemodialysis were associated with MDR.

The most frequently isolated uropathogen in our group was *Escherichia coli*, representing 68.3% of all isolates, similar to other Romanian studies reporting the prevalence of *Escherichia coli* in UTIs ranging from 35.98% to 61.32%, followed by *Klebsiella* spp. 11.2%, in comparison with 17.25% to 22.98% of cases in other groups, *Enterococcus* spp. 6.4%, in comparison with 11.54% to 19.73% in other studies, *Proteus* spp. 6% in our group, ranging from 5.54% to 8.4% in other studies and *Pseudomonas* spp. 6% in our group, ranging from 2.59% to 7.28% in other studies [11,37,38],

In our group, the highest resistance rates were observed to fluoroquinolones at 37.6%, third generation cephalosporines 32.5%, aminoglycosides 34%, aminopenicillins 28.8% and carbapenems 6.1%. Chibelean et al. reported a resistance profile of *Escherichia coli* in UTIs in the Romanian male population, with reported resistance rates of 37.09% to levofloxacin, 28.62% to amoxicillin-clavulanic acid, 8% to Fosfomycin 4.83% to amikacin, and significantly lower resistance to carbapenems—0.4%.

Regarding non-infectious complications in hospitalized patients with UTIs, namely AKI, in a European cohort of 489 CKD patients, the reported rate of AKI in patients hospitalized for UTIs was 73.6%, the mean length of hospital-stay (days) 13.2 ± 18.5 [26] compared to a 34.9% incidence of AKI and 8.33 ± 5.82 days mean length of stay in our group. Other studies reported AKI incidence in hospitalized patients ranging from 45.4% [10] to 75.2% in a kidney transplant cohort [39]. To mention that our group included patients with many comorbidities and risk factors for AKI, both related and non-related to UTIs. In this study we have not analyzed the causality relation between UTI and AKI.

## 5. Conclusions

Chronic kidney disease has a high prevalence among hospitalized patients with urinary tract infections. In addition, those patients are older and with many comorbid conditions. In our study, *Escherichia coli* was the most frequently isolated uropathogen, but *Pseudomonas* spp., *Klebsiella* spp. and *Enterococcus* spp. had the highest antimicrobial resistance profiles. We found that CKD stage G2 to stage G5 is an important risk factor for overall antimicrobial resistance, and even more important, for multiple-drug resistance CKD, increasing the chance of UTIs with MDR strains by 1.779 times, while chronic hemodialysis patients had a 4.068 times higher risk of MDR UTIs. This indicates that AKI is a frequent complication among hospitalized patients with UTIs.

## Figures and Tables

**Figure 1 biomedicines-10-02368-f001:**
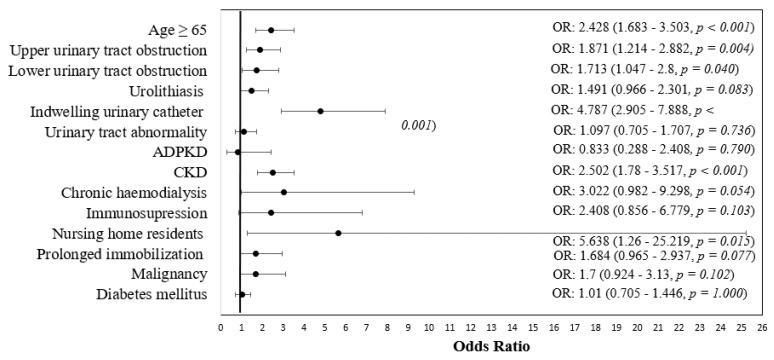
Forest plot for odds ratio with 95% confidence intervals univariate analysis of risk factors for general antibiotic resistance.

**Figure 2 biomedicines-10-02368-f002:**
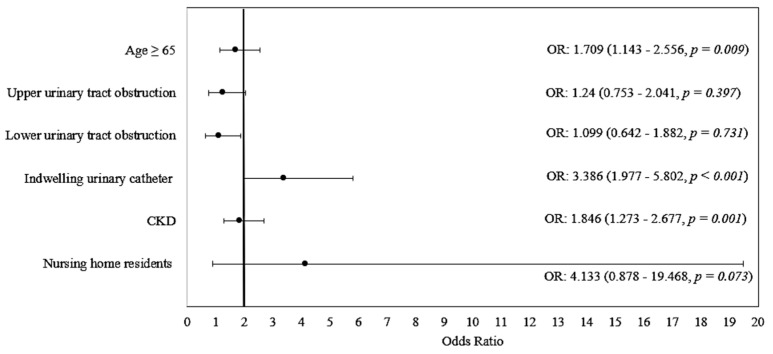
Forest plot for odds ratio with 95% confidence intervals multivariate analysis of risk factors for general antibiotic resistance.

**Figure 3 biomedicines-10-02368-f003:**
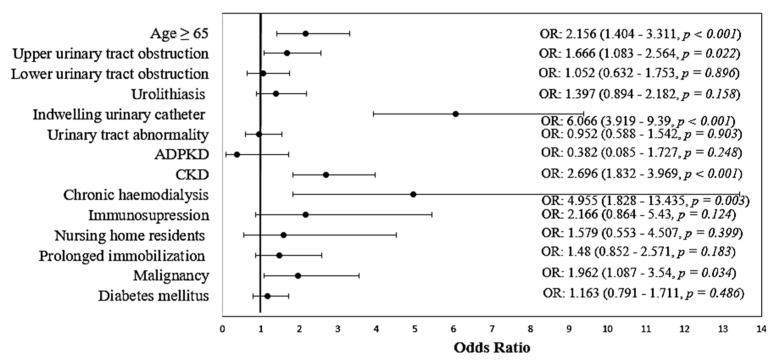
Forest plot for odds ratio with 95% confidence intervals–univariate analysis of risk factors for multi-drug antimicrobial resistance.

**Figure 4 biomedicines-10-02368-f004:**
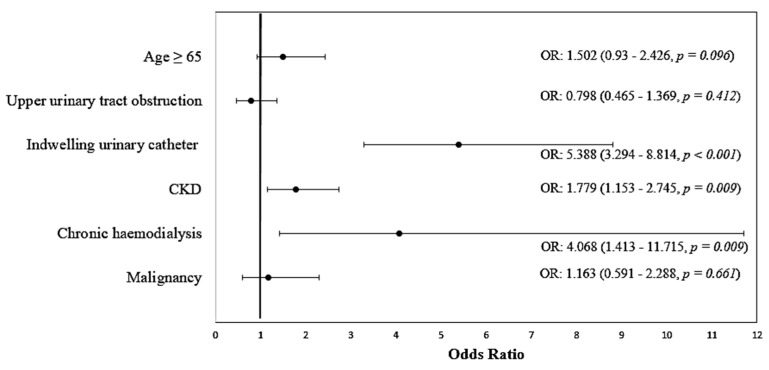
Forest plot for odds ratio with 95% confidence intervals–multivariate analysis of risk factors for multi-drug antimicrobial resistance.

**Table 1 biomedicines-10-02368-t001:** General characteristics of the study patients.

Variable (*n* = 564)	
**Age (Mean ± SD, Median (IQR))**	68.63 ± 17.2, 72 (61–81)
**Age ≥ 65, (*n*, %)**	391 (69.3%)
**Male sex (*n*, %)**	179 (31.8%)
**Infection type (*n* = 562) (*n*, %)**	
**Cystitis**	43 (7.6%)
**Pyelonephritis**	436 (77.4%)
**Urosepsis**	93 (16.5%)
**Recurrent UTIs**	173 (30.7%)
**CAUTI**	72 (12.8%)
**Host-related risk factors for complicated UTIs**	
**CKD**	308 (54.6%)
**Diabetes mellitus**	174 (30.9%)
**Urolithiasis (*n* = 561, NA = 3)**	106 (18.9%)
**Upper urinary tract obstruction**	113 (20.1%)
**Lower urinary tract obstruction**	81 (14.4%)
**Indwelling urinary catheter**	72 (12.8%)
**Ureteral stent**	33 (5.9%)
**Nephrostomy tube**	19 (3.4%)
**Cystostomy tube**	2 (0.4%)
**Pregnancy (*n* = 380)**	9 (2.4%)
**Urinary tract abnormalities (*n* = 563)** **Functional abnormalities** **Congenital malformations** **Acquired anatomical abnormalities**	96 (17.1%) 58 (60.42%) 21 (21.88%) 17 (17.7%)
**ADPKD**	14 (2.5%)
**Immunosuppression**	19 (3.4%)
**Nursing home residents**	15 (2.7%)
**Prolonged immobilization**	61 (10.8%)
**Decubitus ulcer**	20 (3.5%)
**Malignancy**	50 (8.9%)
**Malnutrition**	34 (6%)
**Laboratory data (Mean ± SD, Median (IQR))**	
**WBC count (10^3^/mm^3^) (*n* = 561)**	10 ± 7.18, 9.3 (5.94–14.2)
**CRP (mg/dL) (*n* = 373)**	17.91 ± 47.4, 5.19 (0.75–17.35)
**Fibrinogen (mg/dL) (*n* = 483)**	574.06 ± 198.1, 546 (438–700)
**Serum albumin (g/dL) (*n* = 154)**	3.58 ± 0.92, 3.68 (2.99–4.30)
**Outcomes**	
**Length of hospital stay(days)** **(Mean ± SD, Median (IQR))**	8.33 ± 5.82, 7 (4–10.75)
**All-cause mortality**	42 (7.5%)

**Table 2 biomedicines-10-02368-t002:** Microbial profile of urine cultures according to CKD status.

Etiologic Agent (*n* = 564)	CKD (*n*, %)	Non-CKD (*n*, %)	*p* *
*Escherichia coli*	186 (72.7%)	199 (64.6%)	**0.413**
*Pseudomonas* spp.	12 (4.7%)	22 (7.1%)	
*Staphylococcus* spp.	3 (1.2%)	5 (1.6%)	
*Streptococcus* spp.	2 (0.8%)	2 (0.6%)	
*Proteus* spp.	16 (6.3%)	18 (5.8%)	
*Klebsiella* spp.	22 (8.6%)	41 (13.3%)	
*Enterococcus* spp.	15 (5.9%)	21 (6.8%)	

* Fisher’s Exact Test.

**Table 3 biomedicines-10-02368-t003:** Univariate analysis of risk factors for general antibiotic resistance.

Variable	Sensitive	Resistant	*p* *
**Age ≥ 65**	152 (59.1%)	239 (77.9%)	**<0.001**
**Upper urinary tract obstruction**	38 (14.8%)	75 (24.6%)	**0.004**
**Lower urinary tract obstruction**	28 (10.9%)	53 (17.4%)	**0.040**
**Urolithiasis**	40 (15.6%)	66 (21.6%)	0.083
**Indwelling urinary catheter**	22 (8.6%)	95 (30.9%)	**<0.001**
**Urinary tract abnormality**	42 (16.3%)	54 (17.6%)	0.736
**Autosomal dominant polycystic kidney disease**	7 (2.7%)	7 (2.3%)	0.790
**CKD**	109 (42.4%)	199 (64.8%)	**<0.001**
**CKD stage**			
**KDIGO G1**	5 (4.8%)	1 (0.5%)	**0.021**
**KDIGO G2**	10 (9.5%)	29 (15.1%)	
**KDIGO G3**	47 (44.8%)	66 (34.4%)	
**KDIGO G4**	20 (19%)	54 (28.1%)	
**KDIGO G5**	23 (21.9%)	42 (21.9%)	
**Chronic hemodialysis**	4 (1.6%)	14 (4.6%)	0.054
**Immunosuppression**	5 (1.9%)	14 (4.6%)	0.103
**Nursing home residents**	2 (0.8%)	13 (4.2%)	**0.015**
**Prolonged immobilization**	21 (8.2%)	40 (13%)	0.077
**Malignancy**	17 (6.6%)	33 (10.7%)	0.102
**Diabetes mellitus**	79 (30.7%)	95 (30.9%)	1.000

* Fisher’s Exact Test.

**Table 4 biomedicines-10-02368-t004:** Multivariate logistic regression for general antimicrobial resistance prediction.

Variable	OR (95% CI)	*p*	Model Parameters
**Age ≥ 65**	1.709 (1.143–2.556)	**0.009**	Χ^2^(6) = 75.085 *p* < 0.001 Nagelkerge R^2^ = 0.167 Hosmer–Lemeshow Test: *p* = 0.857 Se: 68.5%, Sp: 61.3% Accuracy: 65.2%
**Upper urinary tract obstruction**	1.240 (0.753–2.041)	0.397	
**Lower urinary tract obstruction**	1.099 (0.642–1.882)	0.731	
**Indwelling urinary catheter**	3.386 (1.977–5.802)	**<0.001**	
**CKD**	1.846 (1.273–2.677)	**0.001**	
**Nursing home residents**	4.133 (0.878–19.468)	0.073	
**Constant**	0.439	**<0.001**	

**Table 5 biomedicines-10-02368-t005:** Distribution of risk factors related to multi-drug antimicrobial resistance.

Variable	Non-MDR	MDR	*p* *
**Age ≥ 65**	256 (64.8%)	135 (79.9%)	**<0.001**
**Upper urinary tract obstruction**	69 (17.6%)	44 (26.2%)	**0.022**
**Lower urinary tract obstruction**	56 (14.2%)	25 (14.9%)	0.896
**Urolithiasis**	68 (17.3%)	38 (22.6%)	0.158
**Indwelling urinary catheter**	44 (11.1%)	73 (43.2%)	**<0.001**
**Urinary tract abnormalities**	68 (17.3%)	28 (16.6%)	0.903
**Autosomal dominant polycystic kidney disease**	12 (3%)	2 (1.2%)	0.248
**CKD**	188 (47.6%)	120 (71%)	**<0.001**
**CKD stage**			
**KDIGO G1**	6 (3.3%)	0 (0%)	**0.037**
**KDIGO G2**	30 (16.5%)	9 (7.8%)	
**KDIGO G3**	69 (37.9%)	44 (38.3%)	
**KDIGO G4**	42 (23.1%)	32 (27.8%)	
**KDIGO G5**	35 (19.2%)	30 (26.1%)	
**Chronic hemodialysis**	6 (1.5%)	12 (7.1%)	**0.003**
**Immunosuppression**	10 (2.5%)	9 (5.3%)	0.124
**Nursing home residents**	9 (2.3%)	6 (3.6%)	0.399
**Prolonged immobilization**	38 (9.6%)	23 (13.6%)	0.183
**Malignancy**	28 (7.1%)	22 (13%)	**0.034**
**Diabetes mellitus**	118 (29.9%)	56 (33.1%)	0.486

* Fisher’s Exact Test.

**Table 6 biomedicines-10-02368-t006:** Multivariate logistic regression for prediction of MDR.

Variable	OR (95% CI)	*p*	Model Parameters
**Age ≥ 65**	1.502 (0.930–2.426)	0.096	Χ^2^(6) = 90.689 *p* < 0.001 Nagelkerge R^2^ = 0.212 Hosmer–Lemeshow Test: *p* = 0.543 Se: 45.8%, Sp: 88.8% Accuracy: 75.9%
**Upper urinary tract obstruction**	0.798 (0.465–1.369)	0.412	
**Indwelling urinary catheter**	5.388 (3.294–8.814)	**<0.001**	
**CKD**	1.779 (1.153–2.745)	**0.009**	
**Chronic hemodialysis**	4.068 (1.413–11.715)	**0.009**	
**Malignancies**	1.163 (0.591–2.288)	0.661	
**Constant**	0.146	**<0.001**	

## Data Availability

The data presented in this study are available on request from the corresponding author. The data are not publicly available due to privacy restrictions.

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
