# Peer review of "Chronic Kidney Disease—An Underestimated Risk Factor for Antimicrobial Resistance in Patients with Urinary Tract Infections"

_biomedicines, 2022, doi:10.3390/biomedicines10102368_

Round 1

Reviewer 1 Report

Vacaroiu et al., have presented here an interesting study on kidney complications due to UTI and drug resistance microbial infections. The sample size of patients (564)in this study is well appreciated. However, there are some minor comments I would like the authors to address.

1. Which antibiotic drugs were found to be the most commonly observed in case of the MDR strains causing UTI?

2. Development of antibiotic resistance is indeed a huge challenge world wide and to tackle this problem as lot of strategies are being trailed. Among which one is the use of nanoparticles particularly of polymer origin such as Alginic acid (https://doi.org/10.1007/s12257-010-0099-7). The author could include this is their discussion as a probable strategy to overcome drug resistance.

3. It is interesting to see the wide range of bacterial strains included in the Table 2 which are known to cause UTI. Are there any bacillus strains also reported for UTI or Kidney infection?

4.In line 247, the word “complication among” has been typed without space.

Author Response

Dear reviewer,

Thank you very much for taking the time to review our paper. We appreciate a lot your efforts and your expertise. We have modified the manuscript according to the comments below.

Point 1: In our group, the highest resistance rates were observed to fluoroquinolones 37.6%, 3rd generation cephalosporines 32.5%, aminoglycosides 34%, aminopenicillins 28.8% and carbapenems 6.1% and we included this information in the revised manuscript. This data was not included it in the initial manuscript because we decided to prepare another paper with extended data related to specific resistance profiles, and for this specific paper we focused the analysis on risk factors for AMR, starting from the point that CKD is an underestimated risk factor for AMR.

Point 2: We read the suggested article with great interest. However, we consider that including this information in discussion we should extend the discussion regarding strategies to reduce AMR and it diverts form the main idea of this paper. But we think it will be a great idea to write a paper focused on strategies to overcome a drug resistance. Please email us if you are interested to write a paper on this subject in collaboration.

Point 3: To our knowledge, there are only a few case reports of UTIs caused by Bacillus genus stains. In our group we had no isolates of Bacillus genus.

Point 4: We corrected the grammatical error

We also added some changes in the introduction and conclusion, according to your statements that this paper sections could be improved.

Reviewer 2 Report

The presented manuscript summarizes interesting observations concerning the phenomenon of antibiotic resistance in people with chronic kidney disease. The manuscript is clear and the statistical calculations are correct. Part of the discussion should highlight the state of the research to date. In the part describing the results, I propose to present them also in a graphic form.

Author Response

Dear reviewer,

Thank you very much for taking the time to review our paper. We appreciate a lot your efforts and your expertise. We have modified the manuscript according to the comments below.

Point 1: We tried to discuss the state of knowledge in the initial manuscript, but we added some completions and to point the importance of our study for the state of knowledge.

Point 2: We included in the revised manuscript forest plot type of graphics for analysis of risk factors for general and multiple drug resistance.

Reviewer 3 Report

Vacaroiu et al showed in their observational study that CKD patients having a 1,846 times higher risk of UTIs with antibiotic resistant strains, while chronic haemodialysis patients had a 4.068 times 185 higher risk of having MDR UTIs. The statistic used is accurate and the study group could include in total 564 adult in-hospital patients. However some questions remain unanswered.

-         The authors should state how pyelonephritis was diagnosed?

-          Was the reason for the AKI only the UTI or did the patient also suffer from other comorbidities (e.g. diabetes, art. hypertension)

-          The authors should add a descriptive analysis 8e.g. co morbidities, how many drugs did the patient took beforehand, how many UTIs did they have beforehand, how often did they had to take antibiotics beforehand) of the patient population

-          The authors should state if the detected bacterial UTI strains are comparable with those described in other hospitals/in the literature. Aare those common strains and are the resistance profile is the same compared to other hospitals in Romania?

Author Response

Dear reviewer,

Thank you very much for taking the time to review our paper. We appreciate a lot your efforts and your expertise. We have modified the manuscript according to the comments below.

Point 1: Pyelonephritis was diagnosed according to EAU Guideline statements, based on clinical presentation: fever >38°C, chills, lumbar pain, vomiting, nausea, costovertebral angle tenderness, with or without cystitis symptoms and imaging finding of pyelonephritis at ultrasound (US) and/or computed tomography (CT) evaluation. We added this information in the reviewed manuscript.

Point 2: In our group we included hospitalized patients with UTIs, thus they are patients with many comorbidities and multiple risk factor for AKI, both related and non-related to UTIs. We did not state in the manuscript if AKI cases were secondary to UTI related factors because we did not collect data about cause/risk factor of AKI. Thus, the finding that AKI is frequent among hospitalized UTI patients is a premise for further causality studies. We added an explanatory statement in this regard.

Point 3: For this study, we collected only data about comorbid conditions known to be risk factors for complicated UTIs or potential risk factors for antimicrobial resistance. We included data about these comorbid conditions and other host related risk factors for complicates UTIs in the general characteristics table. Considering the retrospective character of our study, we could not collect data about previous antibiotic use, previous UTI because this information was missing in many records and we have not included these aspects in our analysis in order to avoid errors.

Note: we already implemented a protocol for a similar prospective study and we included these aspects in the study protocol.

Point 4:   We added in discussion section a paragraph where we compare the detected bacterial UTI strains and resistance profiles with those described in other studies, including in Romanian studies. However, it is important to state that we experience a severe penury of epidemiological studies in Romania and the characteristic of the study populations in other papers published by Romanian colleagues are different from our group.

We also added some changes in the introduction and conclusion, according to your statements that this paper sections could be improved.